# *GhCYP710A1* Participates in Cotton Resistance to Verticillium Wilt by Regulating Stigmasterol Synthesis and Plasma Membrane Stability

**DOI:** 10.3390/ijms23158437

**Published:** 2022-07-29

**Authors:** Li Huang, Guiming Li, Qiaoling Wang, Qian Meng, Fan Xu, Qian Chen, Fang Liu, Yulin Hu, Ming Luo

**Affiliations:** 1Key Laboratory of Biotechnology and Crop Quality Improvement of Ministry of Agriculture/Biotechnology Research Center of Southwest University, Chongqing 400716, China; hl19970914@email.swu.edu.cn (L.H.); lgm5683@163.com (G.L.); wql19980513@163.com (Q.W.); mqhongbin@foxmail.com (Q.M.); xufanfeiren@163.com (F.X.); chenqiansuaige@163.com (Q.C.); lf530805@163.com (F.L.); huyulin2021@163.com (Y.H.); 2Key Laboratory of Horticulture Science for Southern Mountains Regions of Ministry of Education, College of Horticulture and Landscape Architecture, Southwest University, Chongqing 400716, China; 3Academy of Agricultural Sciences of Southwest University, State Cultivation Base of Crop Stress Biology for Southern Mountainous Land of Southwest University, Chongqing 400716, China

**Keywords:** cotton, sterol C22-desaturase, lipid raft, JA, phytosterol, stigmasterol

## Abstract

Cotton is an important economic crop. Cotton Verticillium wilt caused by *Verticillium dahliae* seriously damages production. Phytosterols play roles in plant-pathogen interaction. To explore the function and related mechanism of phytosterols in the interaction between *Verticillium dahliae* and cotton plants, and the resistance to Verticillium wilt, in this study, we analyzed the changes of sterol composition and content in cotton roots infected by *Verticillium dahliae*, and identified the sterol C22-desaturase gene *GhCYP710A1* from upland cotton. Through overexpressing and silencing the gene in cotton plant, and ectopically expressing the gene in *Arabidopsis*, we characterized the changes of sterol composition and the resistance to Verticillium wilt in transgenic plants. The infection of *Verticillium dahliae* resulted in the content of total sterol and each sterol category decreasing in cotton root. The ratio of stigmasterol to sitosterol (St/Si) increased, indicating that the conversion of sitosterol to stigmasterol was activated. Consistently, the expression level of *GhCYP710A1* was upregulated after infection. The GhCYP710A1 has the conservative domain that is essential for sterol C22-desaturase in plant and is highly expressed in root and stem, and its subcellular location is in the endoplasmic reticulum. The ectopic expression of *GhCYP710A1* gene promoted the synthesis of stigmasterol in *Arabidopsis*. The St/Si value is dose-dependent with the expression level of *GhCYP710A1* gene. Meanwhile, the resistance to Verticillium wilt of transgenic *Arabidopsis* increased and the permeability of cell membrane decreased, and the content of ROS decreased after V991 (a strain of *Verticillium dahliae*) infection. Consistently, the resistance to Verticillium wilt significantly increased in the transgenic cotton plants overexpressing *GhCYP710A1*. The membrane permeability and the colonization of V991 strain in transgenic roots were decreased. On the contrary, silencing *GhCYP710A1* resulted in the resistance to Verticillium wilt being decreased. The membrane permeability and the colonization of V991 were increased in cotton roots. The expression change of *GhCYP710A1* and the content alteration of stigmasterol lead to changes in JA signal transduction, hypersensitivity and ROS metabolism in cotton, which might be a cause for regulating the Verticillium wilt resistance of cotton plant. These results indicated that *GhCYP710A1* might be a target gene in cotton resistance breeding.

## 1. Introduction

Verticillium Wilt is a highly destructive vascular plant disease. The pathogens, *Verticillium dahliae* and *Verticillium albo-atrum* belong to the genus of *Verticillium*, and are two main pathogens causing Verticillium Wilt of cotton [1,2,3]. *Verticillium dahliae* has a wide host and infects more than 200 dicotyledon plants. The main crops include cotton, eggplant, sweet pepper, rape, olive, potato, strawberry and tomato [2,4]. Generally, *Verticillium dahliae* can survive for more than 10 years through its resting structure and microsclerotia (MS) in soil and host residues, and can maintain its infectivity to host plants. Therefore, it is very difficult to control cotton Verticillium wilt because of the persistence of MS, the wide host of pathogens, the lack of resistance genes in upland cotton (*Gossypium hirsutum*) and the lack of fungicide [2,5,6,7].

Cotton (*Gossypium* spp.) is the most important fiber crop and oilseed crop, providing 35% of the total fiber used in the world. Cotton is produced in more than 80 countries in the world, of which cotton is the main commercial crop in more than 30 countries. In 2016–2017, the total cotton planting area was about 29.5 million hectares, with a total production of 106.49 million bales. The largest producer of raw cotton is China, followed by India, the United States, Pakistan and Brazil, with output of 30.50, 27.00, 20.90, 8.25 and 6.50 million bales, respectively [8]. Cotton Verticillium wilt caused by *Verticillium dahliae* is a soil-borne disease, which has the characteristics of wide distribution, multiple transmission pathways, long survival time and great destructiveness. Verticillium wilt is one of the most devastating diseases in cotton production and is called the “cancer” of cotton. In China, more than 40% of cotton planting areas are affected by Verticillium wilt, resulting in economic losses of up to USD 250–310 million [9,10]. The symptoms of infected cotton plants are as follows: the leaves turn yellow, wither and fall off, the cotton bolls become smaller and the boll shedding rate is high, resulting in reduced yield and poor quality. Therefore, identifying the genes related to cotton resistance to Verticillium wilt and illuminating their action mechanisms will help to cultivate the germplasm resistant to Verticillium wilt, which has important theoretical significance and application value.

The pathogenesis of Verticillium wilt is still largely unclear. Based on the previous studies, two theories, toxicology and vascular blockage, were proposed. In the theory of vascular blockage, when the plant root is damaged, it will cause a large number of exudates to leak out, and the spores or MS of the pathogen begin to germinate under the stimulation of the exudates to form hyphae, which then invade the root cells of the host plant, and then penetrate into the xylem and vascular tissue. After colonization in the vascular tissue, the hyphae rapidly expand to the aerial part of the plant and the hyphae grow to block the vascular tissue, resulting in the yellowing, necrosis or wilting of the leaves and causing the bolls to fall off, eventually leading to plant death [11,12]. In the theory of toxicology, *Verticillium dahliae* produces toxins that can affect the growth and development of host plants, interfere with the normal metabolism of plants, and lead to the etiolation, necrosis or wilting of leaves of infected plants and the abscission of leaves and cotton bolls [13]. Recently, there has been more evidence to support the toxicology theory. The mycotoxins of *Verticillium dahliae* such as protein-lipopolysaccharide complex and sulfacetamide have been reported [13,14]. Our study revealed that *Verticillium dahliae* produces a sphingosine analogue, fumonisin (FB1), which is a synthesis inhibitor of sphingolipids that is an important component of lipid rafts on membrane [15]. The discovery confirmed that membrane plays an important role in the interaction between host and pathogen. Membrane is a complex structure composed of lipid bilayer and protein, and is responsible for functions in cell activity such as protection, organization, transport and signal transduction [16,17]. Lipid raft is a special structural microdomain in the plasma membrane which is composed of sphingolipids, sterols and proteins [18]. Sterols in lipid rafts mainly include sitosterol, stigmasterol, campesterol, cholesterol, sterol glycoside and acyl sterol glycoside. Sphingolipids mainly include two types of sphingolipids, glycosylceramide (GluCers) and glycosylinositol phosphate ceramide (GIPCs) [19]. It is found that the composition of free and conjugated sterols influences the formation of liquid-ordered phase and liquid-disordered phase of membrane [20,21,22].

Stigmasterol is one of the end products of phytosterol synthesis pathway. Sitosterol is converted to stigmasterol by sterol C22 desaturase (*CYP710As*). Compared with sitosterol, stigmasterol has an unsaturated double bond at C22 of the side chain. Due to the rigidity of the double bond, the flexibility of the alkyl chain is reduced, thus affecting the properties of the lipid bilayer [20,23]. This characteristic makes stigmasterol play an important role in plant-pathogen interaction, which was elucidated by a few reports. The expression level of *AtCYP710A1* was increased in *Arabidopsis* leaves infected with the biotrophic fungus cyclomycetes and the semi-biotrophic fungus oomycetes [24,25]. Disturbing sterol synthesis or declining stigmasterol content increased the plant susceptibility to pathogen. For example, *Arabidopsis* SMT2 (STEROL METHYLTRANSFERASE 2) mutant *smt2* was susceptible to *Pseudomonas syringae* compared with wild-type [26]. Silencing squalene synthase gene (*SQS*) weakened the resistance to *Pseudomonas syringae* and *Xanthomonas campestris* in *Arabidopsis*, and promoted the growth of pathogen *Pseudomonas syringae* by increasing the flow of nutrients into the apoplast [26]. The expression level of *AtCYP710A1* was significantly induced after inoculation with non-host pathogens in *Arabidopsis* and the resistance to *Pseudomonas syringae* reduced in the deletion mutant of *Arabidopsis CYP710A1*. On the contrary, overexpressing *AtCYP710A1* reduced the permeability of cell membrane, limited the release of nutrient components into the apoplast, and thus enhanced the resistance to *Pseudomonas syringae* [26]. Similarly, in *Arabidopsis* with HMGS overexpression, the expression of *AtCYP710A1* was upregulated and stigmasterol content was increased, and resistance to *Botrytis cinerea* and H_2_O_2_ tolerance were enhanced [27]. On the other hand, the pathogenesis-related protein 1 (PR-1) could bind sterols (including stigmasterols) in vitro. PR-1 interferes with the sterol uptake of pathogen (sterol-deficient type) by binding sterols in the environment or directly binding to pathogenic sterols, and then inhibiting the growth of pathogens [28]. These studies revealed that phytosterols, especially stigmasterol, play a role in the resistance to biotic stress. 

It is reported that biotrophic and semi-biotrophic pathogens colonize their hosts by growing in the apoplast and absorbing nutrients from the cytoplasm [29]. It has also been found that pathogen virulence factors can directly bind to the promoters of sugar transporters such as *SWEET11* and *SWEET14* in *Arabidopsis*, influencing sugar content and pathogen growth in the apoplast [30]. Therefore, plants have an unknown mechanism in membrane function to prevent nutrients from flowing into the apoplast. Sitosterol and stigmasterol are important components of plant membrane sterols and play roles that maintain the integrity, fluidity and permeability of cell membrane, so as to improve the stress resistance of plants [23,31,32]. 

In recent years, more and more studies have shown that phytosterols play an important role in plant resistance to biotic and abiotic stresses. However, since phytosterols have many molecular species, it is not clear which sterol molecules play a role in stress resistance and what the related cellular and molecular mechanisms are. In order to understand the function and action mechanism of stigmasterol in cotton resistance to Verticillium wilt, the sterol C22 desaturase gene *GhCYP710A1* that was significantly upregulated by V991 infection was cloned in upland cotton. Through the construction of plant expression vectors, cotton genetic transformation and VIGS analysis, the functions and mechanisms of *GhCYP710A1* and stigmasterol in cotton resistance to Verticillium wilt were clarified, which not only provided experimental data for further elucidating the interaction between cotton and *Verticillium dahliae*, but also provided transgenic cotton plants in which resistance to Verticillium wilt was improved.

## 2. Results

### 2.1. The Effects of Verticillium dahliae Infection on Sterol Content and Gene Expression in Cotton Roots

Phytosterols play important roles in plant-pathogen interaction, but the role of phytosterols in cotton plant-*Verticillium dahliae* interaction is not clear. To explore the function and action mechanism of phytosterols in cotton resistance to Verticillium wilt, the content of phytosterols in cotton roots treated with or without V991 (a strain of *Verticillium dahlia*) spore suspension for 12 h was detected by liquid chromatography tandem mass spectrometry (LC-MS/MS). The results demonstrated that four sterol categories (campesterol, sitosterol, stigmasterol and stigmastanol) and three sterol esters (campesteryl esters, stigmasteryl esters and sitosteryl esters) were detected in cotton roots (Figure 1A). Compared with mock, the content of total sterol and each sterol category decreased in treated cotton root. Given the ratio of various phytosterols, such as the ratio of campesterol to sitosterol (C/S) and the ratio of stigmasterol to sitosterol (St/Si), which play an important role in regulating plant growth and development and plant-pathogen interaction [16,33], we analyzed the changes of C/S and St/Si cotton roots after V991 infection. The results showed that, compared with the control group, the C/S did not change significantly after V991 infection, while the St/Si increased significantly after V991 infection (Figure 1B). These results indicated that the content and composition of plant sterols changed significantly after V991 infection. In the phytosterol biosynthesis pathway, sterol C-22 desaturase (CYP710As) mediates the conversion of sitosterol to stigmasterol [16]. The St/Si increase indicated that the conversion of sitosterol to stigmasterol was activated. Consistently, the expression level of *GhCYP710A1* was upregulated after infection (Figure 1C,D). These results suggested that *GhCYP710A1* might play a role in the cotton resistance to Verticillium wilt.

### 2.2. Characterization of GhCYP710A1 in Upland Cotton

To clone the *GhCYP710A1* gene, based on the cDNA sequence of Gh_D05G1417 in the cotton FGD database (https://cottonfgd.org/ (accessed on 10 September 2019)), we designed two specific primers that recognize the flank sequence of putative ORF. The leaf cDNA of Jimian 14 was used as a template in the PCR system. The resulting sequence was 1509 bp and contained an ORF of 972 bp which encoded 383 amino acid residues. The molecular weight and isoelectric point of the predicted protein was 44.04 kDa and 6.814, respectively. The gene was located on chromosome 5 of the D subgenome of upland cotton (*Gossypium hirsutum* L.). We performed multi-sequence alignment of the amino acid sequence of GhCYP71A1 with the CYP710As from other species. The sequence alignment showed that GhCYP710A1 has 66% homology with SlCYP710A11, 68% with AtCYP710A1 and 65% with AtCYP710A2. The GhCYP710A1 has four transmembrane regions and conserved domain and amino acids of CYP710A enzyme (Figure 2A). There is a conserved sequence of F(L/M)F(A/S)QDA(S/T)(S/T)S in the substrate recognition site (SRS4) of the I helix region of CYP710A protein, which is an important feature of CYP710A protein and is crucial to the activity of C-22 sterol desaturase [34]. The SRS4 domain contains conserved threonine residues (Thr) and alanine residues (Ala). Among which, threonine residues (Thr) participate in the oxygen activation and proton transfer of the hydroxylation reaction of P450 monooxygenase, and alanine residues (Ala) are necessary for the catalytic function of P450 monooxygenase [35]. Phylogenetic relationship analysis showed that GhCYP710A1 is close to HuCYP710A1 (*Herrania umbratica*), TcCYP710A1 (*Theobroma cacao*) and DzCYP710A1 (*Durio zibethinus*), while far from the four AtCYP710As (*Arabidopsis*) (Figure 2B). Taken together, these results suggested that the cloned GhCYP710A1 might be a CYP710A homologue in upland cotton.

### 2.3. Expression Pattern and Subcellular Location of GhCYP710A1

Expression patterns are important data for understanding a gene function. To investigate the expression pattern of *GhCYP710A1* gene in cotton plant, we used qRT-PCR to detect the expression levels of *GhCYP710A1* gene in various cotton organs, tissues and fiber cells. The results showed that *GhCYP710A1* is highly expressed in root, stem, leaf, petal, cotyledon, hypocotyl, pistil and stamen, while it is lowly expressed in ovule and fiber cell (Figure 3A). Considering that stigmasterol, as a “stress” sterol, plays important roles in plant resistance to biotic and abiotic stresses [16], we further investigated the response of *GhCYP710A1* to jasmonic acid (JA) and salicylic acid (SA). The results showed that JA and SA could significantly induce the expression of *GhCYP710A1* (Figure 3B). JA and SA are important plant hormones involved in plant resistance to stress [36], suggesting that *GhCYP710A1* gene might be involved in plant resistance to biotic and abiotic stress. According to the sequence analysis, there were three transmembrane regions in GhCYP710A1 (Figure 2A), suggesting that GhCYP710A1 might be localized to the cell membrane system. To investigate the subcellular localization of GhCYP710A1, we constructed a plant expression vector containing a cassette of CaMV 35S-GhCYP710A1::eGFP-T-NOS (Figure 3C) and then the GhCYP710A1::eGFP and HDEL::mCherry genes were transiently expressed in tobacco leaf. The result indicated that eGFP signal was co-localized with mCherry signal (Figure 3D). Since HDEL is a marker protein localized in endoplasmic reticulum, the result suggests that GhCYP710A1 is localized in endoplasmic reticulum.

### 2.4. Ectopic Expression of GhCYP710A1 Gene Promoted the Synthesis of Stigmasterol in Arabidopsis thaliana

The T-DNA region of constructed plant overexpression vector pLGN-*GhCYP710A1* was transformed into *Arabidopsis thaliana* genome by the flower dipping method. Seven kan-resistance plants were obtained and further confirmed by GUS histochemical staining due to there being a CaMV 35S-GUS-NOS cassette in the T-DNA region (Figure 4A). After 3~4 weeks growth, the gDNA of GUS-positive plants and wild-type (Col-0) was extracted for further identification by PCR. The results showed that the seven plants could amplify a specific band of about 2000 bp, which was similar to that of the positive control (the plasmid of the plant expression vector as a template in PCR system), but no specific band was amplified from the negative control (wild-type gDNA as template in PCR system) and empty control (water as a template in PCR system) (Figure 4B). Taken together, these results indicated that we obtained transgenic *Arabidopsis*. We named the transgenic plants as AtOE-1, AtOE-2, AtOE-3, AtOE-4, AtOE-5, AtOE-6 and AtOE-7 (Figure 4B). To determine the expression of *GhCYP710A1* in transgenic *Arabidopsis* plants, we extracted the total RNA from the leaves of all transgenic plants and wild-type *Arabidopsis* and reverse-transcribed them into the first strand cDNA. By qRT-PCR, we investigated the expression level of *GhCYP710A1* in transgenic plants and wild-type. The results showed that *GhCYP710A1* was successfully expressed in transgenic *Arabidopsis* (Figure 4C). According to the expression level of *GhCYP710A1*, we selected AtOE-2, AtOE-4 and AtOE-7 lines for further study. Among which, AtOE-4 and AtOE-7 were the two lines with the lowest and highest expression, respectively. 

*CYP710A* is a key gene for the conversion of β-sitosterol to stigmasterol. In order to determine the changes of sterol content in transgenic *Arabidopsis* overexpressing *GhCYP710A1*, we detected the sterol content of the leaves of wild-type and transgenic *Arabidopsis* after 3-week growth. The result indicated that, compared with the wild-type, the content of total sterol of all three transgenic plants decreased while the contents of cholesterol, campesterol and sterol ester in all three transgenic plants did not change significantly. The contents of sitosterol and stigmastanol decreased significantly in all transgenic plants while the content of stigmasterol significantly increased by 2.12-, 1.83- and 3.24-fold in AtOE-2, AtOE-4 and AtOE-7 plants, respectively (Figure 4E). Compared with wild-type, the stigmasterol increase showed positive correlation and the sitosterol decrease showed negative correlation with the expression level of *GhCYP710A1* in each transgenic plant (Figure 4E). Furthermore, the ratios of stigmasterol to sitosterol in transgenic plants were significantly higher than those of the wild-type (Figure 4D). These results showed that the *GhCYP710A1* could catalyze the conversion of sitosterol to stigmasterol, suggesting that it is an important gene in regulating the content and proportion of sitosterol to stigmasterol in cotton plants.

### 2.5. Ectopic Expression of GhCYP710A1 Gene Enhanced the Resistance to V991 in Arabidopsis

To investigate the changes of resistance to Verticillium wilt in transgenic *Arabidopsis* overexpressing *GhCYP710A1*, we treated wild-type and transgenic plants with V991 spore suspension (10^7^ conidia/mL). After 10 days of inoculation, there was a significant difference between wild-type and transgenic *Arabidopsis*. The leaves of wild-type *Arabidopsis* had typical symptoms of Verticillium wilt and most of the transgenic plants showed that the leaves were yellowing, wilting or withering, while the disease index of transgenic plants was significantly reduced and only a few plants showed yellowing leaves (Figure 5A). Based on the disease symptoms in the order from weak to strong, we defined five grades that ranged from grade 0 to grade 4. Compared with wild-type, the disease symptom of grade 0 and grade 4 was significantly increased and decreased in three transgenic plants, respectively (Figure 5C). Furthermore, the disease index of wild-type was 74 while those of AtOE-2, AtOE-4 and AtOE-7 were 56, 50 and 44, respectively (Figure 5D). The result indicated that the disease index of transgenic *Arabidopsis* was significantly lower than that of wild-type (Figure 5D). In order to quantify the colonization of *Verticillium dahliae* in *Arabidopsis* leaves, two specific primers ITS1-F and ST-VE1-R for detecting *Verticillium dahliae* were used to investigate the biomass of *Verticillium dahliae* in *Arabidopsis* leaves [37]. The result showed that the relative biomass of *Verticillium dahliae* was about five times lower in transgenic *Arabidopsis* compared with wild-type, indicating that the colonization of *Verticillium dahlia* was significantly lower in transgenic plants than in wild-type (Figure 5D). These results revealed that ectopic expression of *GhCYP710A1* gene could enhance the resistance to Verticillium wilt in *Arabidopsis*. 

Reactive oxygen species (ROS) will accumulate as one of the early stress responses [38]. To investigate the changes of ROS accumulation in wild-type and transgenic plants after being infected by V991, the assay of DAB histochemical staining was carried out. There was no obvious difference between wild-type and transgenic plants before inoculation with V991 (Figure 5B). However, after V991 infection, transgenic leaves displayed a weaker DAB signal compared with wild-type leaves (Figure 5B), indicating less ROS accumulation in transgenic leaves than in wild-type leaves after infection. The result further suggested that ectopic expression of *GhCYP710A1* gene could enhance the resistance to Verticillium wilt in *Arabidopsis*.

### 2.6. The Transgenic Cotton Plants Overexpressing GhCYP710A1 Enhanced the Resistance to V991

Through cotton genetic transformation and transgenic plant identification, transgenic cotton plants overexpressing *GhCYP710A1* gene were obtained (Appendix A). We investigated the resistance of transgenic plants to Verticillium wilt. After inoculation of V991 spore suspension (10^7^ conidia/mL) with cotton root for 15 days, the wild-type plants showed more serious wilting, yellowing and defoliation, while the symptom of disease in transgenic plants was significantly weakened and only partial wilting occurred (Figure 6A). The lesion area in transgenic leaves was obviously smaller than that in control leaves (Figure 6B,E). By calculating the disease grade, the plants that possessed the fourth disease grade accounted for 18% and 18% of the total transgenic cotton plants while the plants that displayed the fourth disease grade accounted for 54% of the total wild-type plants. Contrarily, the transgenic cotton plants that possessed zero and one disease grades accounted for 75% and 76% of the total plants, while the wild-type plants that possessed these disease grades accounted for 40% (Figure 6A,F). Furthermore, the disease index was 60, 28 and 26 in wild-type plants, GhOE-1 plants and GhOE-5 plants, respectively (Figure 6G). The disease indexes (DI) of transgenic cotton plants were far lower than those of control plants (Figure 6G). Further detection found that the biomass of *Verticillium dahliae* in the leaves of transgenic lines was significantly lower than that in control plant leaves (Figure 6H). Subsequently, the same position of the inoculated cotton stem was sectioned and observed. The results showed that the WT vascular tissue was severely browning, while the transgenic lines only had slight browning around the xylem (Figure 6C). Through further observing the growth of V991 labelled by GFP, the result showed that the GFP signal was less in the transgenic root than in the control root (Figure 6D). These results revealed that overexpression of *GhCYP710A1* enhanced the resistance of cotton to Verticillium wilt.

### 2.7. Silencing GhCYP710A1 Reduced Cotton Resistance to V991

Since we did not obtain the stable transgenic cotton plants in which the *GhCYP710A1* expression was suppressed, we downregulated the *GhCYP710A1* expression in cotton plants by VIGS strategy and investigated the changes of resistance to V991 in silencing plants (Appendix A). After inoculation with V991 spore suspension (10^7^ conidia/mL) for 15 days, both TRV:00 and TRV:*GhCYP710A1* plants showed typical symptoms of Verticillium wilt, accompanied by defoliation, necrosis and wilting. However, compared with TRV:00 plants, the disease of TRV:*GhCYP710A1* plants was more serious, and some plants died (Figure 7A). By calculating the disease grade, the silencing *GhCYP710A1* plants with disease grade 0 and 1 accounted for 54% of the total plants while plants with disease grade 4 accounted for 9% of the total investigated plants. Contrarily, the control plants (TRV:00) with disease grade 0 and 1 accounted for 29% of the total plants while plants with disease grade 4 accounted for 5% of the total investigated plants (Figure 7A,F). The disease index of TRV:*GhCYP710A1* plants was far higher than that of TRV:00 plants (Figure 7G). Furthermore, the lesion area in silencing *GhCYP710A1* transgenic leaves was obviously bigger than that in control leaves (Figure 7C,E). Subsequently, the sections at the same position of cotton stems were observed. After 20-day inoculation with V991 strain, the transverse and longitudinal sections of the stems of TRV:00 plants and TRV:*GhCYP710A1* plants were browning. Compared with TRV:00 plants, the vascular tissue of TRV:*GhCYP710A1* plants browned more deeply and the cell necrosis was more serious (Figure 7B). The fungus-specific primers ITS1-F and ST-VE1-R were used to detect the biomass of V991 strain in cotton leaves. The results showed that the biomass of V991 in the leaves of silencing *GhCYP710A1* plants was almost five times that of TRV:00 plants (Figure 7H). To further detect the colonization of V991 in TRV:00 and silencing of *GhCYP710A1* plants, we conducted the recovery growth experiment of fungus on the stems of TRV:00 plants and silencing plants after 20-day inoculation. The result showed that the growth of fungal colonies in the stems of TRV:*GhCYP710A1* plants was more than that of the control plants, indicating that the colonization degree of V991 in TRV:*GhCYP710A1* plants was greater (Figure 7D). These results indicated that silencing the *GhCYP710A1* gene could decrease the cotton resistance to Verticillium wilt.

### 2.8. Modifying GhCYP710A1 Expression Resulted in the Alteration of Plasma Membrane Permeability and ROS Accumulation in Transgenic Cotton Plants

After inoculation with V991 spore suspension (10^7^ conidia/mL) for 15 days, the signals of DAB and trypan blue were weaker in transgenic leaves than in control leaves (Figure 8A,B). These results indicated that overexpressing GhCYP710A1 inhibited ROS accumulation and cell death in cotton leaves after infection with V991 strain. Previous studies reported that ROS caused lipid peroxidation in cell membrane and led to cell membrane damage, and destroyed the permeability and integrity of cell membrane [39]. Therefore, we detected the relative conductivity of cotton leaves infected by V991. The results showed that the relative conductivity of leaves of transgenic lines was significantly lower than that of control leaves (Figure 8C), indicating that the permeability of cell membrane of transgenic cotton leaves was reduced and the degree of cell damage was light.

By the same protocol, we investigated the changes on the silencing *GhCYP710A1* plants. The results showed that the signals of DAB and trypan blue were stronger in silencing *GhCYP710A1* leaves than in control leaves (Figure 8D,E), indicating that silencing *GhCYP710A1* enhanced ROS accumulation and cell death in cotton leaves after infection with V991 strain. 

Furthermore, the relative conductivity of the leaves of the silencing *GhCYP710A1* plants was significantly higher than that of TRV:00 plants (control) after being inoculated with Verticillium wilt (Figure 8F), indicating that silencing the *GhCYP710A1* gene increased the membrane permeability of cotton leaves after V991 infection. 

Previous studies reported that the composition of free sterols and conjugated sterols influenced plasma membrane feature such as the liquid-ordered phase and liquid-disordered phase of plasma membrane [20]. To understand the effect of changes in stigmasterol content on the plasma membrane features, we observed the di-4ANEPPDHQ-labeled cotton roots with a confocal microscope. The results showed that the GP value of transgenic cotton roots was significantly higher than that of the control root (Figure 8G,H). The result showed that overexpression of the *GhCYP710A1* gene could enhance the order of plasma membrane and improve the stability of plasma membrane.

### 2.9. Modifying GhCYP710A1 Expression Led to the Expression Changes of Disease Resistance Genes

To explore the resistance mechanism of transgenic cotton overexpressing *GhCYP710A1* to Verticillium wilt, we detected the expression of genes involved in disease resistance in the cotton root infected by V991 for 12 h. The selected genes were mainly involved in JA biosynthesis (AOS, LOX1 and OPR3); signal-response-related genes (JAZ1 and JAZ3); SA-signal-pathway-related genes (NDR1, NPR1 and PR1); marker genes of hypersensitivity pathway (HIN1 and HSR203); genes related to active oxygen scavenging (POD, SOD and CAT); and genes related to production (RbohB). The results showed that the expression of AOS, LOX1, JAZ1, HIN1, HSR203, POD, SOD and CAT was significantly higher in transgenic cotton overexpressing *GhCYP710A1* than in control after infection by Verticillium wilt, while the expression of NDR1, NPR1 and PR1 in transgenic cotton had no significant change compared with that in wild-type (Figure 9A). The results showed that overexpression of *GhCYP710A1* could activate JA signal pathway, hypersensitivity and ROS signal pathway, but did not affect SA signal pathway.

Contrarily, the expression levels of LOX1, OPR3, JAZ1, JAZ3, HIN1, SOD, POD and CAT were significantly lower in silencing *GhCYP710A1* plants than in TRV:00 plants, while the expression of RbohB was significantly higher in silencing *GhCYP710A1* plants (Figure 9B), indicating that silencing the *GhCYP710A1* gene inhibited JA signal pathway, hypersensitivity and ROS accumulation, but did not affect SA signal pathway. Taken together, modifying *GhCYP710A1* expression led to the expression changes of disease resistance genes.

## 3. Discussion

### 3.1. GhCYP710A1 and Stigmasterol Positively Regulated Cotton Resistance to Verticillium Wilt

Phytosterols are an important component of biomembranes. Phytosterols and sphingolipids mainly present in lipid raft, a functional region of the membrane which mediates the integrity, fluidity and permeability of the membrane and participates in regulating the activity of membrane-binding proteins, sensing environmental conditions, and regulating stress resistance and signal transduction [31,32,40,41]. As important bioactive molecules, phytosterols are not only the precursor for the synthesis of the important plant hormone brassinolide, but are also important regulators of plant growth activities such as seed germination, plant phenotype, cellulose synthesis, lignin deposition and cell wall formation [42,43,44,45,46]. Recently, a few studies reported that sterols play an important role in plant resistance to biotic and abiotic stresses [16]. *Pseudomonas syringae* is a plant pathogen that causes many plant diseases [47]. Overexpressing *AtCYP710A1* enhanced the resistance to *Pseudomonas syringae* in *Arabidopsis*. On the contrary, *AtCYP710A1* deletion mutant reduced the resistance to *Pseudomonas syringae* [26]. After inoculation with *Pseudomonas syringae* and *Botrytis cinerea*, the transcription level of *AtCYP710A1* (At4g05320) and the content of stigmasterol were significantly increased in *Arabidopsis* leaves, and the ratio of stigmasterol to β-sitosterol was elevated in cell membrane [48]. Consistently, the expression level of *AtCYP710A1* was upregulated in *Arabidopsis* leaves infected with the living trophic fungus cyclomycetes and the semi-living trophic fungus oomycetes [24,25]. These results were similar to the result from cotton root infected by *Verticillium dahliae*, indicating that CYP710A1 plays an important role in the interaction between plants and pathogens. In the transgenic *Arabidopsis* plants overexpressing hydroxymethyl glutaryl CoA synthase (HMGS), the *AtCYP710A1* transcript and stigmasterol were upregulated, which enhanced the resistance of *Arabidopsis* to *Botrytis cinerea* [27]. In our study, after inoculation with *Verticillium dahliae*, the phytosterol content in cotton roots decreased, but the expression of *GhCYP710A1* increased, resulting in an increase in the ratio of stigmasterol to sitosterol (St/Si) in cotton roots. This suggested that the content and proportion of stigmasterol and sitosterol might play a role in the interaction between cotton plant and *Verticillium dahliae*. Ectopic expression of *GhCYP710A1* promoted the conversion of sitosterol to stigmasterol in transgenic *Arabidopsis*, resulting in the decrease of sitosterol content and the increase of stigmasterol content. Consequently, the St/Si value increased in transgenic *Arabidopsis* and the resistance of transgenic *Arabidopsis* to *Verticillium dahliae* increased. Consistently, overexpression of *GhCYP710A1* enhanced the resistance to *Verticillium dahliae* in transgenic cotton plants. On the contrary, silencing the *GhCYP710A1* gene by VIGS decreased cotton plant resistance to *Verticillium dahliae*. These results revealed that the *GhCYP710A1* expression level and the content of its product stigmasterol play a positive role in the regulation of cotton resistance to Verticillium wilt.

### 3.2. Effect of Membrane Lipid Changes on the Infection Process of Verticillium dahliae

Biomembrane is an important site of plant cells to sense pathogen invasion and resist pathogens [49]. More and more studies indicated that lipid raft is a key region of membrane, which can not only affect the integrity, fluidity and permeability of the membrane, but can also sense and transmit the infection signal of pathogenic bacteria and start the immune response of plant cells [31,32,40,41]. In order to study the possible changes of membrane feature in the early response of plant defense signal transduction, tobacco cells were treated with cryptococcin and bacterial pathogen flagellin (two effectors that induce plant defense response). The result indicated that the proportion of liquid-ordered phase in plasma membrane increased temporarily and the fluidity of membrane increased [50]. Sterols and sphingolipids are two important components of membrane which are concentrated in the lipid raft region. The content and composition of sphingolipids and the content and composition of free sterols and conjugated sterol can affect the formation of liquid-ordered phase in membrane [20]. The increase of stigmasterol content can maintain the fluidity and permeability of cell membrane, restrict the release of nutrients into the apoplast, and enhance the resistance to *Pseudomonas syringae* and *Botrytis cinerea* [26,27]. In this study, it was found that the relative conductivity of transgenic cotton leaves overexpressing *GhCYP710A1* after inoculation with Verticillium wilt was significantly lower than that of wild-type, indicating that the permeability of cell membrane was reduced. On the contrary, the relative conductivity of leaves of silencing *GhCYP710A1* plants was significantly higher than that of wild-type plants, indicating that the plant cell membrane was seriously damaged and the permeability of cell membrane was enhanced. Our results also showed that the growth of *Verticillium dahliae* strain labeled by eGFP (V991-eGFP) was inhibited in the root of transgenic cotton overexpressing *GhCYP710A1*, which was further confirmed by the biomass of V991 being reduced in transgenic leaves. Previous studies reported that biotrophic or semi-biotrophic pathogens could colonize their hosts by growing in apoplast and obtaining nutrients from the cytoplasm [29]. Our results indicated that overproduction of stigmasterol improved the stability of the membrane, which restricted the release of nutrients to the apoplast and inhibited the colonization of *Verticillium dahliae* in the root tissue, thus improving the cotton resistance to Verticillium wilt.

### 3.3. GhCYP710A1 Regulates Cotton Resistance to Verticillium Wilt by Regulating ROS, HR and JA Signaling Pathways

Previous reports have shown that ROS (such as H_2_O_2_) will accumulate when plants are infected with *Verticillium dahliae* [51]. Lower-level ROS was involved in cross-linking of cell wall components, regulating related gene expression, signal transduction of hypersensitivity reaction (HR) and killing invasive pathogens [52,53,54,55]. HR could not only inhibit the growth of pathogens, but could also stimulate specific defense responses of adjacent tissues and systemic acquired resistance of plants [56]. However, high-level ROS causes toxicity to plant cells and oxidative damage. Therefore, plants form complex regulatory mechanisms to maintain the dynamic balance of ROS, such as antioxidant systems (CAT, POD and SOD) and reactive oxygen species generation systems (RbohA and RbohB) [57,58]. In this study, DAB staining showed that ROS accumulation in transgenic *Arabidopsis* and transgenic cotton leaves overexpressing *GhCYP710A1* was significantly reduced compared with the control plants. Consistently, the expression levels of genes involved in ROS scavenging system (CAT, POD and SOD) and HR marker genes (HIN1 and HSR203) were significantly upregulated; on the contrary, ROS accumulation increased significantly in the leaves of silencing *GhCYP710A1* plants, which might result from downregulating the expression of CAT, POD and SOD and upregulating RbohB.

Plant hormones play an important regulatory role in plant immunity [59]. SA (salicylic acid) and JA (jasmonic acid) are two key hormones in plant resistance to pathogens. SA signaling is mainly triggered by biotrophic pathogens and participates in systemic acquired resistance (SAR), while JA signaling is mainly activated by necrotrophic pathogens and participates in local acquired resistance (LAR) [60]. Furthermore, the accumulation of SA and JA is often associated with plant HR [61,62]. Griebel and Zeier reported that stigmasterol synthesis was induced by flagellin, polysaccharide and ROS, while it did not depend on the signaling of SA, JA and ethylene [48]. This result differed from our results. *GhCYP710A1* expression was significantly upregulated by SA and JA. Furthermore, JA signaling was activated in transgenic cotton plants overexpressing *GhCYP710A1*, while it was suppressed in silencing *GhCYP710A1* plants, indicating *GhCYP710A1* expression and stigmasterol synthesis have a close relationship with SA and JA signaling. This difference might be caused by different pathogens. *Verticillium dahliae* is a fungus while *Pseudomonas syringae* was used in the study of Griebel and Zeier.

## 4. Materials and Methods

### 4.1. Plant and Verticillium dahliae Materials and Growth Conditions

*Arabidopsis* plants were cultivated in a greenhouse at 21 °C for 16 h light period and at 19 °C for 8 h dark period at 72% relative humidity. In the greenhouse, supplemental light (100 Wm^−2^) was used when the sunlight influx intensity was below 150 Wm^−2^.

The wild-type cotton cultivar Jimian 14 (*Gossypium hirsutum* L.) was provided by Professor Ma Zhiying (Hebei Agricultural University, China). The transgenic plants were cultivated at the Biotechnology Research Center of Southwest University. All plants were grown under natural conditions in the experimental field of the Biotechnology Research Center of Southwest University in Chongqing, China. 

According to whether *Verticillium dahliae* can cause the symptoms of cotton leaf abscission or not, the strains can be divided into defoliation and non-defoliation type [63,64]. The defoliation type may be lethal to plants while non-defoliation type only causes slightly wilted leaves and does not wither leaves, and the plants infected by non-defoliation type strain can finally return to normal growth [63]. *Verticillium dahliae* strain V991 is a defoliation type with high virulence. The V991 was preserved by the Biotechnology Research Center of Southwest University, and the strain used for inoculation was grown in potato dextrose broth (PDB) or Chawton’s medium (CZA) at 26 °C.

### 4.2. Subcellular Localization of GhCYP710A1

The coding sequences of *GhCYP710A1* were amplified and cloned into the binary vector pCAMBIA2300-GFP to construct the GhCYP710A1-GFP vectors. The constructed vectors were transformed into *Agrobacterium tumefaciens* strain GV3101, then transiently expressed in *tobacco* leaves as reported [65]. Transfected *tobacco* plants were kept in the dark for 16 h and then moved to light conditions for 48 h. The fluorescence signals were detected using confocal laser scanning microscopy (Leica, Wetzlar SP8, Germany). HDEL was used for endoplasmic reticulum markers. All experiments were performed with three independent biological replicates.

### 4.3. Vectors Construction and Transformation

The *GhCYP710A1* gene was amplified from cDNA of cotton roots using the *GhCYP710A1*-specific primers, and the resulting fragment was used to construct the plant expression vector pLGN (containing two expression cassettes, CaMV 35S promoter::GhCYP710A1::nos and 35S promoter::GUS:NPTII::nos). The resulting vector was introduced into *Agrobacterium tumefaciens* strain LBA4404, which was used to transform upland cotton (*Gossypium hirsutum* L.) as previously described [33]. To silence *GhCYP710A1*, the silencing fragment was amplified from the cotton genome DNA using the *GhCYP710A1* VIGS primers, and homologous recombination was conducted to integrate the sequence into pTRV2 using a Ligation-Free Cloning Kit (ABM, Vancouver, Canada) to construct the *GhCYP710A1* VIGS vector. The resulting vector was introduced into *Agrobacterium* strain GV3101. The *GhCYP710A1* VIGS strain was mixed with the strain containing pTRV1 and injected into the cotyledon of Jimian 14 seedlings to generate *GhCYP710A1* VIGS plants. 

### 4.4. RNA Extraction and qRT-PCR Assay

Total RNA was extracted from the samples using a Plant Total RNA Extraction kit (Tiangen, Beijing, China). First-strand cDNA was synthesized from 1 µg total RNA using a Reverse Transcription Kit with Genomic DNA Remover (Takara, Kusatsu, Japan). Real-time quantitative PCR was performed on a CFX96™ Optical Reaction Module (Bio-Rad, Hercules, CA, USA) using Novostar-SYBR Supermix (Novoprotein, Suzhou, China) according to the manufacturer’s instructions. GhHISTONE3 was used as an internal control. Each analysis was repeated with three biological replicates.

### 4.5. Investigation of Plant Resistance to Verticillium dahliae

To investigate plant resistance to *Verticillium dahliae*, the plants were grown for four weeks. The conidial suspension of *Verticillium dahliae* (1 × 10^9^ conidia/L) was irrigated to the roots of the plants and cultured for 15 days, and then Verticillium wilt symptoms were recorded. To detect the *Verticillium dahliae* growth on plant leaves, TRV:00 and TRV:GhCYP710A1 leaves at the same developmental stage were collected. The leaf surface was wounded and 5 µL spore solution was introduced into each wounded site. The lesion area was measured with ImageJ software (http://rsbweb.nih.gov/ij/, accessed on 10 August 2021) after 10 days of infection. At least eight lesions were measured in each experiment, and the experiment was repeated at least three times. To quantify fungal biomass, the leaves of three inoculated plants were harvested after 7 days inoculation. qRT-PCR was performed with two specific primers, ITS1-F, AAAGTTTTAATGGTTCGCTAAGA and ST-VE1-R, CTTGGTCATTTAGAGGAAGTAA [37]. GhHISTONE3 was used as an internal control. Each analysis was repeated with three biological replicates. For histological observation of mycelial growth, the stems of three inoculated plants were collected at 15 days after inoculation and cut longitudinally. The samples were photographed under a SYCOP3 stereoscope (Zeiss, Oberkochen, Germany). 

For the colonization of dahlias on cotton roots after dahlias treat cotton roots, this study used the spore suspension (10^6^ conidia/L) of V991-GFP (*Verticillium dahliae*-GFP) to infect cotton roots for 5 days. The fluorescence signals were detected using confocal laser scanning microscopy (Leica, Wetzlar SP8, Germany).

### 4.6. Trypan Blue Staining

After inoculation with *Verticillium dahliae* for 15 days, leaves from the TRV:00 and TRV:GhCYP710A1 plants were stained by boiling in lactophenol trypan blue and subsequently destained with chloral hydrate. The leaves were photographed with a light microscope (Leica Microsystems TCS SP2 AOBS, Wetzlar, Germany).

### 4.7. Reactive Oxygen Species (ROS) Detection in Cotton Leaves

After the cotton seedlings were inoculated with *Verticillium dahliae* for 15 days, cotton leaves were collected and washed using distilled water. The production and accumulation of ROS species were detected using the 3′,3′-Diaminobenzidine (DAB) staining. An optical microscope (Nikon, Tokyo, Japan) was used to observe and photograph the stained ROS patches.

### 4.8. Verticillium Dahliae Recovery Assay

To investigate the growth of *Verticillium dahliae* in the infected plants, the stem fragments (1.0 cm) of the first internode in the seedlings after being infected for 20 days were cultured on potato dextrose broth (PDB) medium at 25 °C. Each experiment was carried out for at least three biological replicates.

### 4.9. Di-4-ANEPPDHQ Stain and Observation

Di-4-ANEPPDHQ was purchased from Invitrogen (CAT #D36802). The methods of Di-4-ANEPPDHQ stain, observation and GP calculation followed the previous report [66]. 

### 4.10. GUS Histochemical Staining

Leaves from transgenic plants were examined for the presence of the CaMV 35S promoter-NPTII::GUS-nos cassette in T-DNA region. The plant leaves were incubated overnight at 37 °C in the staining solution (1.5 mg/mL 5-Bromo-4-chloro-3-indolyl-beta-D-glucuronide, 50 mM sodium phosphate buffer, pH 7.0, 0.1% (*v*/*v*) Triton X100, 0.5 mM potassium ferricyanide and 0.5 mM potassium ferrocyanide). After incubation, the plant leaves were faded in a clearing solution of 8:3:1 (*w*/*v*/*v*) chloral hydrate: distilled water: glycerol. Then, the samples were observed by stereoscope (SteREO Discovery, V20, Zeiss, Gottingen, Germany).

### 4.11. Determination of Relative Conductivity

After the leaves were washed with deionized water and dried, round leaves of the same size were taken with a hole puncher and treated in ultrapure water for 24 h. A conductivity meter was used to detect the conductivity (R1) of the solution, it was boiled in a boiling water bath at 100 °C for 20 min, and after cooling to room temperature, the conductivity (R2) of the solution was detected again. According to the calculation formula: relative conductivity (%) = (R1/R2) × 100% to calculate the relative conductivity.

### 4.12. Sample Collection, Lipid Extraction and Lipidomics

The roots of ten inoculated Jimian 14 were collected after *Verticillium dahliae* inoculation for 12 h. The aerial parts of wild-type (Col-0) and transgenic *Arabidopsis* grown for 3 weeks were collected and immediately put into liquid nitrogen, and were then kept at −80 °C.

After the sample collection, lipid extraction and lipidomic analysis were performed by the Lipidall Technologies Company Limited (http://www.lipidall.com/ (accessed on 10 July 2021)) as described previously [67]. 

### 4.13. Statistical Data Analysis

The software used for statistical data analysis is GraphPad Prism 8.3 and the software used for phylogenetic tree analysis is MEGA5.2. Data were presented as mean ± SD. Statistical data analysis was performed by the one-tailed Student’s *t*-test. *, ** and *** indicate significant differences at *p* < 0.05, <0.01 and <0.001, respectively.

## 5. Conclusions

In this study, we first characterized the profile of phytosterols in cotton root and their changes when responding to the infection of *Verticillium dahliae*. Based on the changes, consequently, we identified the sterol C22 desaturase gene (*GhCYP710A1*) from upland cotton which contributed to the conversion of sitosterol to stigmasterol. Overexpression of *GhCYP710A1* enhances the resistance to Verticillium wilt either in *Arabidopsis* or in upland cotton while the silencing *GhCYP710A1* gene decreases the cotton resistance to Verticillium wilt. The mechanism of the resistance alteration was further investigated to a certain degree. These results suggest that *GhCYP710A1* might be a target gene in cotton resistance breeding. Given that phytosterols are important components in the lipid raft of membrane, future study might focus on the effect of sterol change on the signal transduction of biotic stress since some pathogen elicitor receptors are localized on the lipid raft.

## Figures and Tables

**Figure 1 ijms-23-08437-f001:**
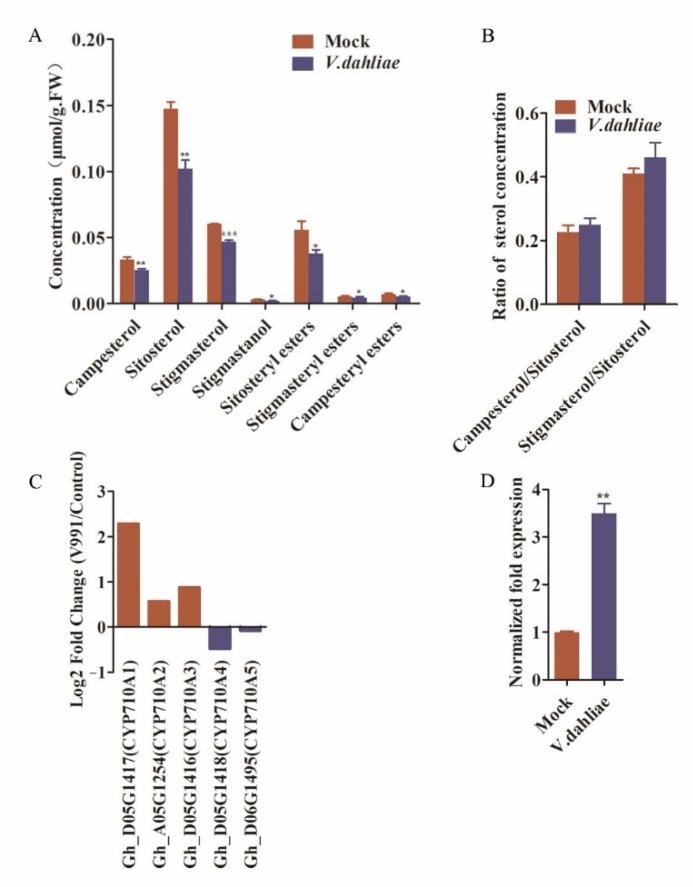
The changes of sterol content and *GhCYP710As* expression in cotton root after *Verticillium dahliae* inoculation. (**A**) The content changes of phytosterols. (**B**) The changes of the ratio of campesterol to sitosterol and stigmasterol to sitosterol. (**C**) The change fold of expression level in cotton root after *Verticillium dahliae* inoculation, which is derived from RNA-Seq. (**D**) The expression level of *GhCYP710A1* was induced in cotton root after *Verticillium dahliae* inoculation. * represents *p* < 0.05, ** represents *p* < 0.01, *** represents *p* < 0.001.

**Figure 2 ijms-23-08437-f002:**
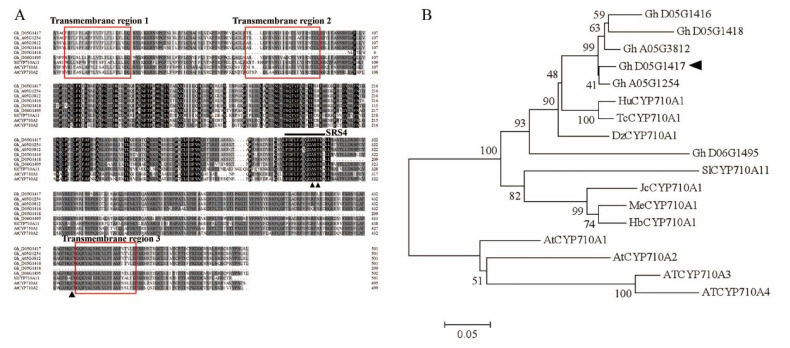
Multi-sequence alignment of GhCYP710A1 with other species CYP710A proteins and phylogenetic tree analysis of CYP710A homologues. (**A**) Multi-sequence alignment of GhCYP710As and SlCYP7010A11 (*Solanum lycopersicum*) and AtCYP710As (*Arabidopsis thaliana*). Black shading represents conserved amino acid residues; gray shading represents similar amino acid residues; the black line indicates conserved amino acids of the substrate recognition site SRS4 in the I-helix region; ▲ indicates the conserved alanine residues (A), threonine residues (T) and heme ligand cysteine residue (C). Red box indicates the transmembrane region. (**B**) Phylogenetic analysis was performed using the MEGA5.2 software with the neighbor-joining (NJ) method under 1000 replicates of bootstrap. ◀ indicates GhCYP710A1. HuCYP710A1 (*Herrania umbratica*, accession number is XP_021294269); TcCYP710A1 (*Theobroma cacao*, accession number is EOY27725); DzCYP710A1 (*Durio zibethinus*, accession number is XP_022777277); SlCYP710A11 (*Solanum lycopersicum*, accession number is NP_001234514); MeCYP710A1 (*Manihot esculenta*, accession number is XP_021621148); HbCYP710A1 (*Hevea brasiliensis*, accession number is XP_021647622); AtCYP710A1~4 (*Arabidopsis thaliana*, with accession numbers NP_180997, NP_180996, NP_180451 and NP_180452, respectively).

**Figure 3 ijms-23-08437-f003:**
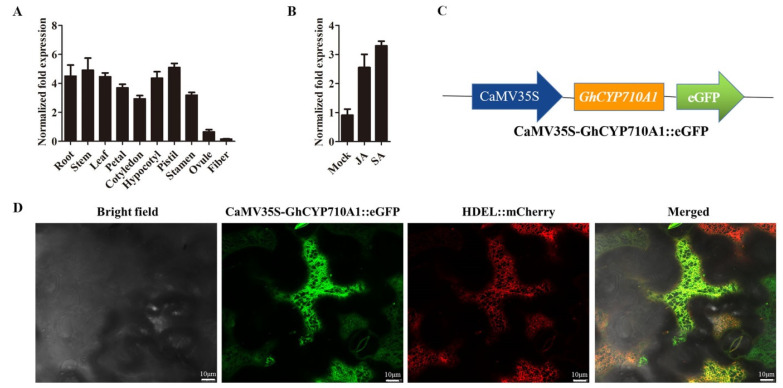
The expression patterns and subcellular localization of GhCYP710A1. (**A**) The expression patterns of GhCYP710A1 in various tissues and organs. Ovule (with fiber cells) was taken from 0-DPA ovary; fiber was taken from 10-DPA ovary. (**B**) The expression of GhCYP710A1 responded to jasmonic acid (JA) and salicylic acid (SA). Mock: the ovule (with fiber cells) grew on the BT medium; JA: the ovule (with fiber cells) grew on the BT medium containing JA; SA: the ovule (with fiber cells) grew on the BT medium containing SA. Error bars represent standard deviation (SD) of three independent replicates. (**C**) Diagram of the CaMV35S-GhCYP710A1::eGFP-T-NOS cassette. (**D**) Subcellular localizations of GhCYP710A1; HDEL::mCherry was used as a positive control indicating endoplasmic reticulum localization. Merge: The merged images of Bright Field, CaMV35S-GhCYP710A1::eGFP and HDEL::mCherry; scale bar = 10 μm.

**Figure 4 ijms-23-08437-f004:**
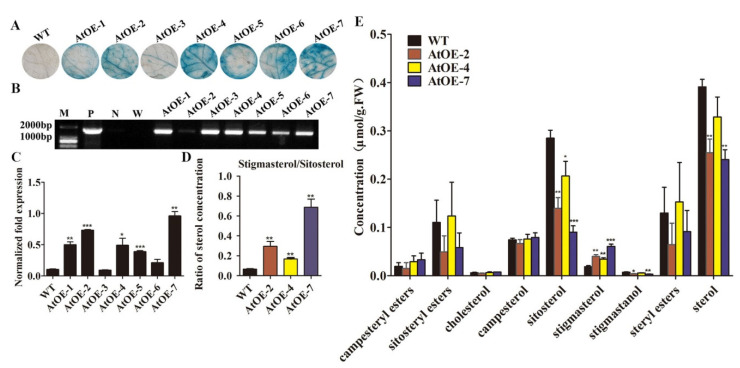
Identification of ectopic expression of GhCYP710A1 gene in *Arabidopsis* and the change of sterol profiles in transgenic plants and control plants. (**A**) GUS histochemical identification of transgenic *Arabidopsis*. (**B**) Molecular identification of transgenic *Arabidopsis*. Two primers that recognized CaMV 35S promoter and the 3′-sequence of *GhCYP710A1* were used in PCR system. M: DNA marker; P: positive control using the pLGN-GhCYP710A1plasmid as template; N: negative control using wild-type *Arabidopsis* gDNA as template; W: empty control using water as template; AtOE-1~AtOE-7: transgenic *Arabidopsis* plants 1# to 7#. (**C**) Relative expression level of GhCYP710A1 in the transgenic *Arabidopsis*. (**D**) The ratio of stigmasterol to sitosterol. (**E**) The content of various free sterols and sterol esters in wild-type and transgenic *Arabidopsis*. Control: wild-type *Arabidopsis* (Col-0); AtOE-2, AtOE-4 and AtOE-7: transgenic *Arabidopsis* line 2#, 4# and 7#, respectively. Error bars represent the standard deviation of three independent replicates. * represents *p* < 0.05, ** represents *p* < 0.01, *** represents *p* < 0.001.

**Figure 5 ijms-23-08437-f005:**
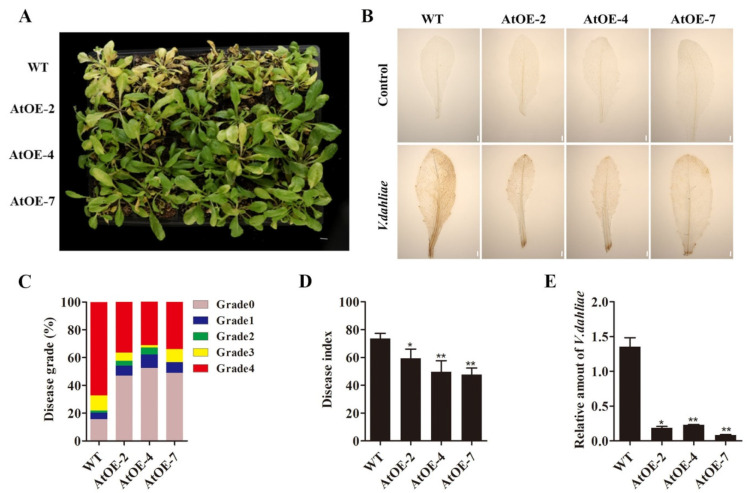
Ectopic expression of *GhCYP710A1* gene in *Arabidopsis* with enhanced resistance to *Verticillium*
*dahliae*. (**A**) Symptoms of *Arabidopsis thaliana* 10 days after Verticillium wilt infection. WT: wild-type *Arabidopsis* control; AtOE-2, AtOE-4, AtOE-7: transgenic *Arabidopsis* lines. Scale bar = 1 cm. (**B**) DAB staining of wild-type and transgenic *Arabidopsis* leaves after *Verticillium dahliae* infection for 10 days. The brown material reflects the hydrogen peroxide content. Scale bar = 1 mm. (**C**) Statistics of disease grade. (**D**) Statistics of disease index. The resistance of transgenic *Arabidopsis* was evaluated by disease index. (**E**) qRT-PCR was utilized to analyze fungal biomass in wild-type and transgenic *Arabidopsis* leaves after *Verticillium dahliae* infection for 10 days. DNA content of *Verticillium dahliae* internal transcribed spacer (ITS) was used as a fungal biomass standard; DNA levels of *Arabidopsis* actin were used as an internal reference. Error bars represent SD of three independent replicates. * represents *p* < 0.05, ** represents *p* < 0.01.

**Figure 6 ijms-23-08437-f006:**
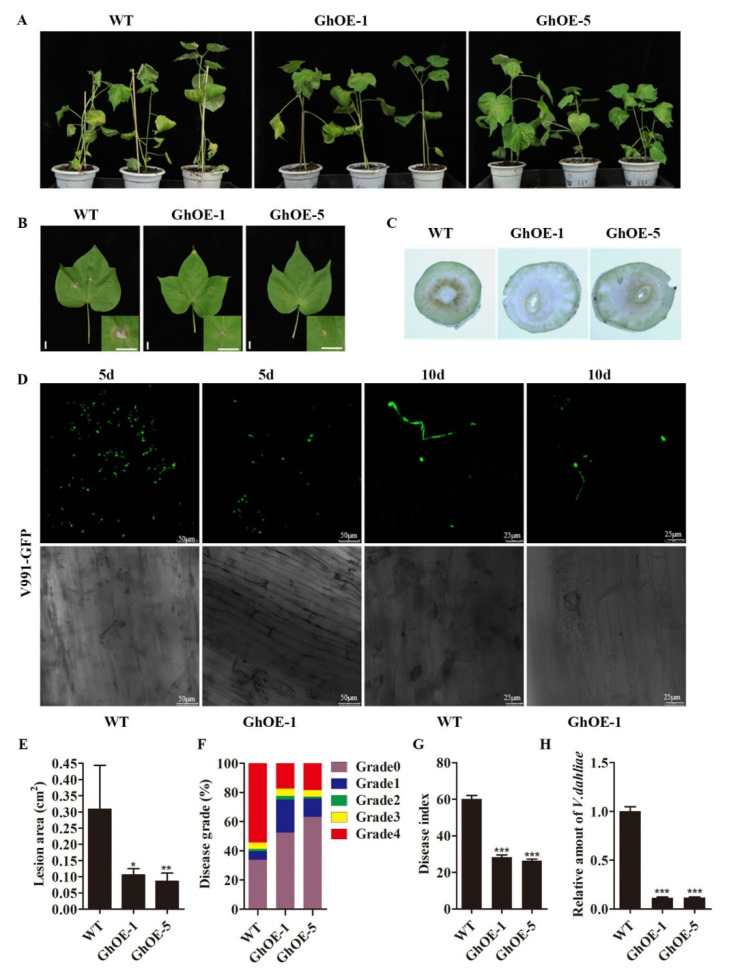
*GhCYP710A1* overexpression in cotton with enhanced resistance to *Verticillium dahlia*. (**A**) Symptoms of cotton after *Verticillium dahliae* infection for 15 days. WT: wild-type cotton; GhOE-1 and GhOE-5: transgenic cotton lines. (**B**) Symptoms of cotton leaves after *Verticillium dahliae* infection for 5 days. Scale bar= 1 cm. (**C**) The cross-sections of cotton stem after *Verticillium dahliae* infection for 15 days; the brown area indicates the diseased vascular tissue. (**D**) The taproots of cotton seedlings with GFP fluorescence were observed under confocal microscope. WT: wild-type cotton; GhOE-1: *GhCYP710A1*-overexpression transgenic cotton lines. (**E**) The lesions area statistics of cotton leaves after *Verticillium*
*dahliae* infection. (**F**) Disease grade statistics. (**G**) Disease index statistics. (**H**) qRT-PCR was utilized to analyze fungal biomass in wild-type and transgenic cotton leaves after *Verticillium dahliae* infection for 15 days. DNA content of *Verticillium dahliae* internal transcribed spacer (ITS) was used as a fungal biomass standard; DNA levels of cotton His were used as an internal reference. Error bars represent the standard deviation of three independent replicates. * represents *p* < 0.05, ** represents *p* < 0.01, *** represents *p* < 0.001.

**Figure 7 ijms-23-08437-f007:**
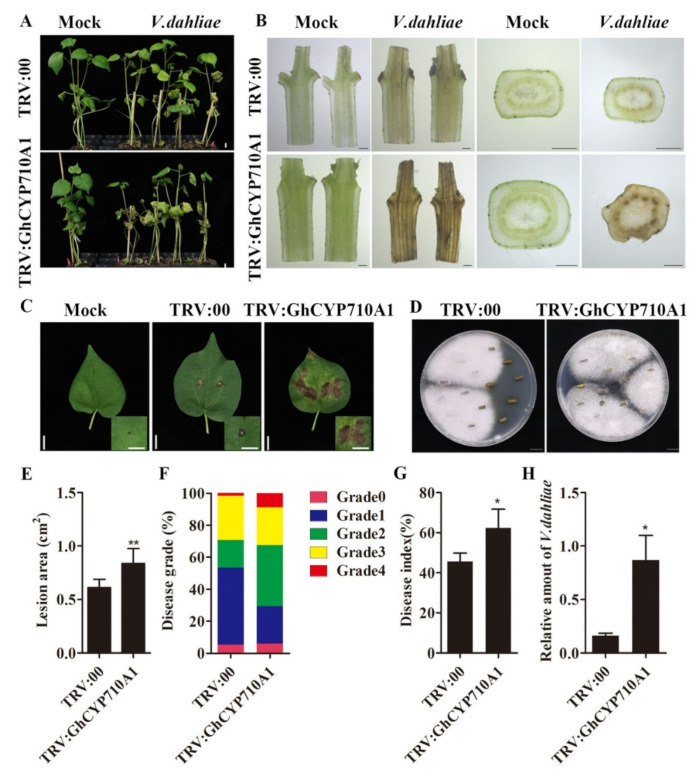
Silencing *GhCYP710A1* enhances the sensitivity of plants to *Verticillium dahliae*. (**A**) Disease symptoms of TRV:00 and TRV:GhCYP710A plants after *Verticillium dahliae* infection for 15 days. Scale bar = 1 cm. (**B**) The cross- and longitudinal sections of cotton plant stems after *Verticillium dahliae* infection for 20 days. The brown area represents the diseased vascular tissue. Scale bar = 1 mm. (**C**) Symptoms of cotton leaves after *Verticillium dahliae* infection for 10 days. Scale bar = 1 cm. (**D**) Stem sections at 20 dpi were plated on potato dextrose agar medium and incubated at 25 °C. Photographs were taken 10 days after culture. Each independent experiment contains 10 plants per treatment. Scale bar = 1 cm. (**E**) The lesions area statistics of cotton after *Verticillium dahliae* infection. (**F**) Disease grade of cotton plants after *Verticillium dahliae* infection for 15 days. (**G**) Disease index of cotton plants after *Verticillium dahliae* infection for 15 days. (**H**) The relative amount of *Verticillium*
*dahliae* determined by qRT-PCR in TRV:00 and TRV:GhCYP710A1 plants after *Verticillium dahliae* infection for 15 days. * represents *p* < 0.05, ** represents *p* < 0.01.

**Figure 8 ijms-23-08437-f008:**
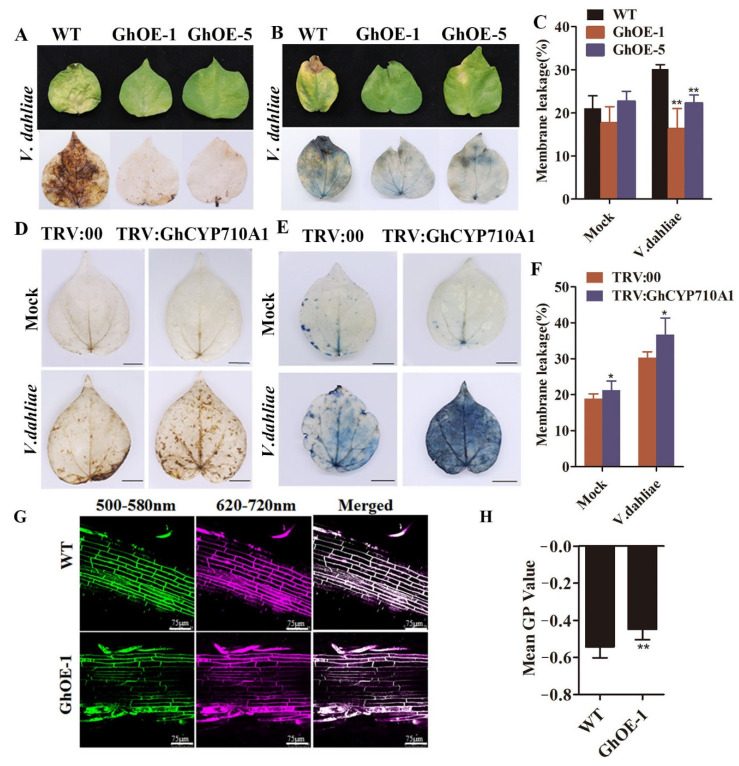
Overexpression of *GhCYP710A1* inhibits ROS accumulation and cell membrane damage in cotton. Silencing of *GhCYP710A1* leads to ROS accumulation and cell membrane damage in cotton. (**A**) DAB staining of wild-type and transgenic cotton leaves after *Verticillium dahliae* infection for 15 days. The brown material reflects the hydrogen peroxide content. WT: wild-type cotton; GhOE-1 and GhOE-5: *GhCYP710A1*-overexpression transgenic cotton lines. (**B**) Trypan blue staining for the *Verticillium dahliae* infection of wild-type and *GhCYP710A1*-overexpression transgenic cotton. (**C**) Electrolyte leakage in the leaves of wild-type and *GhCYP710A1*-overexpression transgenic cotton. ** represents *p* < 0.01. (**D**) The accumulation of ROS in TRV:00 and TRV:GhCYP710A1 plant leaves by DAB staining after *Verticillium dahliae* infection for 15 days. (**E**) Trypan blue staining of TRV:00 and TRV:GhCYP710A1 cotton after *Verticillium dahliae* infection for 10 days. (**F**) Electrolyte leakage in the leaves of TRV:00 and TRV:GhCYP710A1 cotton. Scale bar = 1 cm. * represents *p* < 0.05. (**G**) Fluorescence imaging of di-4-ANEPPDHQ-labeled cotton root cells. WT: wild-type cotton; GhOE-1: *GhCYP710A1*-overexpression transgenic cotton; 500–580 nm: the green channel; 620–720 nm: the red channel; Merged: the merged channel of green and red channels. (**H**) The GP of wild-type and transgenic cotton root cells. ** represents *p* < 0.01.

**Figure 9 ijms-23-08437-f009:**
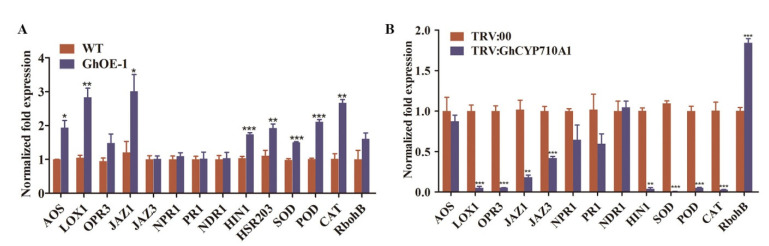
The expression changes of genes involved in disease resistance after *Verticillium dahliae* infection. (**A**) The expression levels of JA-/SA-mediated genes, HR-associated genes and ROS-related genes in root tissue of wild-type cotton and transgenic cotton by qRT-PCR after *Verticillium dahliae* infection for 12 h. WT: wild-type cotton; GhOE-1: *GhCYP710A1*-overexpression transgenic cotton. (**B**) The expression levels of JA-/SA-mediated genes, HR-associated genes and ROS-related genes in TRV:00 and TRV:*GhCYP710A1* cotton leaves by qRT-PCR after *Verticillium dahliae* infection for 7 days. Error bars represent the standard deviation of three independent replicates. * represents *p* < 0.05, ** represents *p* < 0.01, *** represents *p* < 0.001.

## Data Availability

Not applicable.

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
