# Peer review of "GhCYP710A1 Participates in Cotton Resistance to Verticillium Wilt by Regulating Stigmasterol Synthesis and Plasma Membrane Stability"

_ijms, 2022, doi:10.3390/ijms23158437_

Round 1

Reviewer 1 Report

Taken together, the experiment results and presentation indicate Verticillium dahliae infection in cotton is complex. The gene identified to be important in regulating cotton resistance to Verticillium wilt could be one piece of the puzzle in developing resistant cultivars (correctly concluded that it "...might be a target gene in cotton resistance breeding.").

Kindly describe (defoliating, non-defoliating?) type and source of Verticillium dahliae strain V991. While results were promising, they are not definitive, and it will be helpful to understand characterization and origin of V991.

In 'Results' section text, it is not necessary, and somewhat distracting, to bring percentage number to two decimal places of accuracy. One or none is sufficient for this type of discussion, and should be applied consistently.

Review the document and be certain Verticillium wilt and Verticillium dahliae are used properly, properly capitalized, and/or italicized.

Nice work.

Author Response

We are submitting our revised manuscript for your consideration of publication on International Journal of Molecular Sciences. According to reviewers’ great suggestions, we have carefully checked and made some changes. We appreciate all your suggestions and hope this revised manuscript has fully addressed your concerns. Thanks for your consideration. The point-to-point response is detailed as follows.

Best regards,

Sincerely,

Ming Luo

Key Laboratory of Biotechnology and Crop Quality Improvement, Ministry of Agriculture/Biotechnology Research Center, Southwest University.

Chongqing 400716, China

Email: luo0424@126.com

Response to reviewer 1

Question 1: Taken together, the experiment results and presentation indicate Verticillium dahliae infection in cotton is complex. The gene identified to be important in regulating cotton resistance to Verticillium wilt could be one piece of the puzzle in developing resistant cultivars (correctly concluded that it "...might be a target gene in cotton resistance breeding.").

Answer:Thank you very much for your good suggestion. Based on your advice, we have used "...might be a target gene in cotton resistance breeding".

Question 2:Kindly describe (defoliating, non-defoliating?) type and source of Verticillium dahliae strain V991. While results were promising, they are not definitive, and it will be helpful to understand characterization and origin of V991.

Answer:Thank you very much for your good suggestion. We have added type and source of Verticillium dahliae strain V991 to the Materials and methods.

According to whether Verticillium dahliae can cause the symptoms of cotton leaf abscission or not, the strains can be divided into defoliation and non-defoliation type (Jiménez-Díaz et al., 2011, Hu et al., 2015). The defoliation type may be lethal to plants while non-defoliation type only causes slightly wilt leaves and does not wither, and the plants infected by non-defoliation type strain can finally return to normal growth (Jiménez-Díaz et al., 2011). Verticillium dahliae strain V991 is a defoliation type.

HU, X. P., GURUNG, S., SHORT, D. P. G., SANDOYA, G. V., SHANG, W. J., HAYES, R. J., DAVIS, R. M. & SUBBARAO, K. V. 2015. Nondefoliating and Defoliating Strains from Cotton Correlate with Races 1 and 2 of Verticillium dahliae. Plant Dis, 99, 1713-1720.

JIMéNEZ-DíAZ, R. M., OLIVARES-GARCíA, C., LANDA, B. B., DEL MAR JIMéNEZ-GASCO, M. & NAVAS-CORTéS, J. A. 2011. Region-wide analysis of genetic diversity in Verticillium dahliae populations infecting olive in southern Spain and agricultural factors influencing the distribution and prevalence of vegetative compatibility groups and pathotypes. Phytopathology, 101, 304-15.

Question 3:In ' Results ' section text, it is not necessary, and somewhat distracting, to bring percentage number to two decimal places of accuracy. One or none is sufficient for this type of discussion, and should be applied consistently.

Answer:Thank you very much for your good suggestion. Based on your suggestion, we have checked and revised all percentage numbers.

Question 4:Review the document and be certain Verticillium wilt and Verticillium dahliae are used properly, properly capitalized, and/or italicized.

Answer:Thank you very much for your suggestion. We examined the use of Verticillium dahliae and Verticillium wilt in the documentation and made changes based on your suggestions, including capitalization and italics.

Reviewer 2 Report

The paper is well written and structured. The results are clearly explained and the discussion well written.

I have just a few suggestions to improve the final result:

INTRODUCTION:

-line 52: please add some references

-line 56: please add some information about the commercial and economical importance of cotton

RESULTS:

-Figure 1 and all the figures: please  if possible, enlarge the figures or divide them into several separate figures; it is very difficult to analyze and look at them in detail.

- lines 320-330: is this paragraph really necessary? The subject of ROS is described in detail below

CONCLUSIONS

Please rewrite the conclusions. At the moment they are just the repetition of the results. Please add the relevance of this study if some parts of it are new (they are being described for the first time) and suggest other studies that could be undertaken with the results obtained by this work.

Author Response

We are submitting our revised manuscript for your consideration of publication on International Journal of Molecular Sciences. According to reviewers’ great suggestions, we have carefully checked and made some changes. We appreciate all your suggestions and hope this revised manuscript has fully addressed your concerns. Thanks for your consideration. The point-to-point response is detailed as follows.

Best regards,

Sincerely,

Ming Luo

Key Laboratory of Biotechnology and Crop Quality Improvement, Ministry of Agriculture/Biotechnology Research Center, Southwest University.

Chongqing 400716, China

Email: luo0424@126.com

Response to reviewer 2

Question 1:INTRODUCTION:-line 52: please add some references

Answer:Thank you very much for your suggestion. Based on your suggestion, we have added three references to the introduction on line 52

Question 2:INTRODUCTION: -line 56: please add some information about the commercial and economical importance of cotton

Answer:Thank you very much for your suggestion. Based on your advice, we have added detail information about the commercial and economical importance of cotton to the introduction.

Question 3:RESULTS: -Figure 1 and all the figures: please if possible, enlarge the figures or divide them into several separate figures; it is very difficult to analyze and look at them in detail.

Answer:Thank you very much for your suggestion. After receiving your valuable comments, we have carefully checked all the pictures in the article and found that the size and definition of the pictures are sufficient. It may be that the pictures have been dragged and dropped in the document, resulting in changes in the picture clarity. We will add original images of all images in the submission system.

Question 4:RESULTS:- lines 320-330: is this paragraph really necessary? The subject of ROS is described in detail below.

Answer:Thank you very much for your valuable suggestion. Based on your suggestion, we have rewritten this paragraph.    

Question 5: CONCLUSIONS: Please rewrite the conclusions. At the moment they are just the repetition of the results. Please add the relevance of this study if some parts of it are new (they are being described for the first time) and suggest other studies that could be undertaken with the results obtained by this work.

Answer:Thank you very much for your comment. We have rewritten the conclusion.
